# Knowledge, Awareness, and Prevention of Norovirus Infection among Kindergarten Parents in Chengdu, China

**DOI:** 10.3390/ijerph19031570

**Published:** 2022-01-29

**Authors:** Yuanyuan Li, Wenyu Cheng

**Affiliations:** 1School of Education and Science, Sichuan Normal University, Chengdu 610066, China; Yuanyuanli1992@163.com; 2College of Education, Ankang University, Ankang 725000, China; 3School of Modern Agriculture and Biotechnology, Ankang University, Ankang 725000, China

**Keywords:** human noroviruses, kindergarten parents, children, knowledge, awareness, prevention

## Abstract

Human noroviruses (HuNoVs) are a leading cause of acute gastroenteritis among children in China. However, little is known about parents’ knowledge of HuNoV infection and their understanding of how to prevent and control the disease. Therefore, we performed an exploratory survey to assess the level of knowledge of HuNoV infection among kindergarten parents. A cross-sectional survey was conducted by investigating kindergarteners’ parents through an online self-administered questionnaire between October 2020 and November 2020 in Chengdu, China. A total of 771 questionnaires were received with valid responses, and 81.97% of respondents had heard about NoV before. Among parents who had heard about HuNoV before, they had a poor awareness of incubation period, duration, and high-incidence seasons of HuNoV infection. The respondents also had a low-level awareness of how to clean the places contaminated by vomitus or stool. The multiple-regression analysis confirmed that factors associated with good knowledge regarding HuNoV infection were level of education, occupation, history of infection, and HuNoV learning experience. The most expected approach to learn about HuNoV among parents was the internet, followed by knowledge training in kindergartens, community information, and television. This is the first study to assess kindergarten parents’ knowledge and awareness of HuNoV infection. The survey results provide insights that would help in developing effective strategies and educational materials to prevent and control the disease.

## 1. Introduction

Human noroviruses (HuNoVs), previously known as Norwalk viruses, some of the most common causative agents of viral gastroenteritis, are responsible for approximately 20% of cases of acute gastroenteritis [1,2]. The results of HuNoV infection mainly affect children under five years of age, causing an estimated 200,000 child deaths and billions of hospitalizations annually around the world and imposing a heavy societal burden on the economy and public health [2,3]. HuNoV is highly infectious, as only a few infectious particles can cause illness. Symptomatic cases of HuNoV infection include self-limiting diarrhea, nausea and vomiting, as well as abdominal pain, chills, and fever, normally persisting for 2–4 days after onset [4,5]. Notably, the symptoms of the infection can be more severe and prolonged, with a higher risk for mortality in the elderly, the immunocompromised, and children below two years of age [4,5]. The high numbers of viral particles in vomitus and stool, together with high resistance to environmental conditions of the virus and direct person-to-person contact transmission, make HuNoV highly contagious and transmittable [6,7]. HuNoV outbreaks generally occur in any places of public gathering, such as kindergartens, schools, hospitals, prisons, military settings, and swimming pools, where secondary transmissions often occur, which contributes to HuNoV being so prevalent [7]. Currently, no approved vaccine is available to prevent HuNoV infection; thus, the prevention of virus transmission may be the most effective way to reduce the risk of HuNoV infection [8].

Generally, HuNoV is transmitted through the fecal–oral route and infects people who eat food or water contaminated by infectious vomitus or stool [9]. Most commonly, the initial case of HuNoV infection occurs from intaking contaminated food or water, and person-to-person spread further results in an outbreak [9]. HuNoVs are highly diverse and have been divided into seven different genogroups with at least 34 genotypes [10]. Moreover, new emerging variants occur with particular frequency in genotype GII.4, the most prevalent HuNoV [10,11,12]. According to the pandemics of NoV gastroenteritis in the past decades, new variants of GII.4 have appeared on average every 2–3 years [13,14]. Consequently, most of the individuals show no long-lasting immunity to norovirus, and repeated infections may occur throughout a person’s life [12,15].

Since 2006, outbreaks of NoV gastroenteritis have continuously increased and emerged as a major public health concern in China. In a retrospective study conducted by Zhou et al. based on the literature published before 2017, the results showed that the pooled prevalence of HuNoV infection in all ages was nearly 20% nationwide, with two prevalence peaks at the ages of 2 and 40 years old, of which the proportions of NoV gastroenteritis were 23.5% and 32.4%, respectively [16]. Wang et al. performed a nationwide review to estimate the positive rate of HuNoV infection in acute diarrhea cases associated with viral pathogens based on the data retrieved from the active surveillance system of China’s CDC between 2009 and 2018 [17]. Of the 77,855 cases tested for NoV, 12.47% of the patients (9707/77,855) were positive for HuNoV, with an infection peak at ages below 5 years old [17]. Besides this, other epidemiological surveys of NoV gastroenteritis also revealed high positive rates in children under 5 years old, with a significant disease and economic burden of the infection [11,18]. Only in Chengdu, the capital city of Sichuan province, 128 outbreaks of HuNoV infection associated to 1275 patients occurred in schools and kindergartens in 2018 [19]. Given the continued outbreaks of NoV gastroenteritis and the unavailability of a vaccine, preventive measures are critical to the control of the disease. In 2015, China’s CDC issued the Norovirus Outbreak Management and Disease Prevention Guidelines (2015 version). The guidelines on the prevention and control strategies include case management, hand hygiene, environmental disinfection, food and water safety, risk assessment, and health education. However, the NoV outbreaks have not decreased in the past few years. Several studies have attempted to evaluate the perception of NoV gastroenteritis among infection preventionists, undergraduates, and food handlers [20,21,22]. At present, information on the perception of NoV infection among parents of kindergarten children, who are among the highly susceptible age groups, is scant. Therefore, the current study attempts to evaluate the awareness and knowledge about NoV gastroenteritis among parents of kindergarten children in Chengdu, China.

## 2. Materials and Methods

### 2.1. Study Setting and Participants

During October and November 2020, a stratified, random cluster-sampling method was used to conduct a cross-sectional online survey in urban areas of Chengdu city, Sichuan province. Participants were parents whose children attended nine different kindergartens, including three government-owned kindergartens and six privately-owned kindergartens. For the selected kindergartens, two classes from each grade (younger class: mean age = 39.12 months, range: 26–49 months; middle age class: mean age = 55.23 months, range: 50–61 months; and older class: mean age = 67.54 months, range: 62–73 months) were randomly selected, and a total of 54 classes conducted the survey. One of the children’s parents (father/mother), the one who was the most involved in child feeding, was invited to participate in the study with the task of completing a self-administered questionnaire. A total of 810 parents were invited to participate in the study. Undoubtedly, the anonymity of participants was ensured.

This study was reviewed and approved by the Ethical Committee of Sichuan Normal University (Chengdu, China; permit no. 2020101201). The participants were informed about the purpose, benefits, and significance related to this research study on the front page of the questionnaire. Moreover, the participants indicated their consent and agreement to participate by clicking the starting button.

### 2.2. Questionnaire

The self-administered questionnaire used in this study was developed based on the Norovirus Outbreak Management and Disease Prevention Guidelines (2015 version) issued by China’s CDC and partly adapted from previously validated tools [20,21,22,23]. In total, 30 questions covering 4 sections were included in the questionnaire, namely sociodemographic characteristics, pathogenic knowledge of HuNoV infection, attitudes toward disease prevention and control, and assessments of knowledge requirements. The first section, sociodemographic characteristics, is comprised of 6 items, including education background of participants, occupation, grades of children, types of kindergartens (i.e., government-owned, privately-owned), history of children’s HuNoV infection, and HuNoV learning experience. The second section included 11 questions and assessed the pathogenic knowledge of HuNoV, with only one correct answer for single-choice questions. In the section regarding the attitudes toward disease prevention and control, the participants were asked 10 yes/no and single-answer questions to assess the practice behavior towards HuNoV infection. When participants answered these 21 questions, one point was gained after the correct option of each question was checked. Otherwise, zero points were counted. Accordingly, the total scores of Section 2 and Section 3 were 11 and 10, respectively. The parents’ knowledge levels were defined as “good” or “poor” based on an 80% cutoff value of the scores [24]. For example, with a total score of 11 in Section 2, a participant securing 9 or more was categorized as “good” and otherwise was defined as “poor.” The last part focused on investigating knowledge requirements in participants, including requirements of knowledge learning, the source of knowledge regarding HuNoV, and the way to learn knowledge in future. A standalone question was posed and set between the first section and the second section of the questionnaire to ask participants “Have you ever heard about norovirus/ Norwalk virus before?” If the responses were “Yes,” participants were asked to complete all the questions. Otherwise, the questions of the second section, third section, and partial items of the fourth section were concealed without asking for the participants’ responses.

To ensure the reliability and validity of the questionnaire, experts in China’s CDC and doctors in two kindergartens were invited to give advice and revise the questionnaire. Afterwards, a preliminary survey was administered to parents of two kindergartens before the final investigation. Ninety-six valid questionnaires were collected and subjected to evaluation of their reliability and validity using the Cronbach’s α and the Kaiser–Meyer–Olkin (KMO) value. The overall Cronbach’s α coefficient was 0.77, and the Kaiser–Meyer–Olkin validity statistic reached 0.71, showing good reliability and validity of the questionnaire. Finally, the self-administered questionnaire was distributed in electronic form using an online tool named “Wen Juan Xing” for data collection.

### 2.3. Quality Control

Due to the outbreak of COVID-19, this survey was conducted online by sending the questionnaire link to the parents related to the selected classes. Participants who had not participated previously in a similar survey met the inclusion criteria of this investigation. Only one of the children’s parents (father/mother) in each family was invited to complete the questionnaire, and each participant could only submit the filled-out questionnaire when all items were answered.

### 2.4. Data Analysis

The data from the questionnaire were downloaded into Excel 2010 (Microsoft, Redmond, WA, USA). Invalid data or missing data were excluded, and then, all valid data were imported into SPSS version 25.0 software (SPSS, Chicago, IL, USA) for the analyses. Descriptive statistics, including frequency and percentage, were calculated on the sociodemographic characteristics of survey respondents. The differences in sociodemographic statuses were compared with the knowledge scores using an independent *t*-test or a one–way analysis of variance. Logistics linear regression models were performed to assess factors that affected HuNoV infection knowledge among responders. A multivariate analysis was performed based on the significantly independent variables from the bivariate analysis (*p* ≤ 0.2). *p*-values of <0.05 were considered to indicate a statistical significance.

## 3. Results

Of the 810 questionnaires distributed, 771 (95.19%) were received with valid responses. Among the 771 parents, 632 (81.97%) had heard about NoV before. Therefore, those (*n* = 632) who had heard about NoV were subjected to an evaluation of their knowledge score, while the analyses of the sociodemographic characteristics and knowledge requirements used responses of all 771 parents.

### 3.1. Sociodemographic Characteristics of Participants

In this survey, more than half of the children’s parents (*n* = 485, 62.91%) were from privately-owned kindergartens, and 286 (37.09%) were government-owned kindergarten children’s parents. According to the response of the 771 parents, 31.91% (*n* = 246) of their children were in the junior class, 39.43% (*n* = 304) were in the middle class, and 28.66% (*n* = 221) were in the top class. A total of 176 parents had a graduate degree, and 353 parents graduated with a bachelor’s degree, whereas 166 and 76 parents had only completed their vocational degree education and high school degree and below, respectively. Of 771 respondents, 36 were medical staff, and 158 were teachers, while the remaining 577 parents were engaged in other jobs.

### 3.2. Knowledge about HuNoV Infection

Among those (*n* = 632) who had heard about HuNoV before, 3.79% (*n* = 24) of their children had a history of HuNoV infection. According to their responses, the mean total knowledge score for HuNoV infection was only 16.23 ± 2.78. As presented in Table 1, the statistical difference in the scores of parents with different education backgrounds was significant (*p* < 0.001), which showed that parents with bachelor’s degrees and graduate degrees obtained higher scores than those who had a vocational degree as well as those with a high school degree and below educational background. The results from the one-way analysis of variance showed that the knowledge scores differed significantly in terms of occupation (*p* < 0.001) and HuNoV learning experience (*p* < 0.001) of the parents. There was a significant difference in the scores of pathogenic knowledges of HuNoV among parents whether their children had an infection history or not (*p* < 0.05), while no statistical significance was found in the scores of disease prevention and control. The statistical difference in the grades of children or types of kindergartens was not significant.

Next, we assessed the cognition level among parents based on their responses to 21 questions (Table 2). Only 15 parents were able to correctly answer all of the 21 knowledge questions. The average awareness rate of parents was only 62.64%, suggesting a low level of awareness of HuNoV infection. Parents showed a higher awareness rate for disease prevention and control (63.22%) than pathogenic knowledge of HuNoV (62.12%). Only 74.97% parents knew that HuNoV is infectious. They had a low-level awareness of incubation period, duration, and high-incidence seasons of HuNoV infection. Most of the respondents correctly identified the high-risk outbreak places, susceptible population, and route of disease transmission for HuNoV infection; however, nearly 40% did not know that a person is probably infected with HuNoV more than once. Regarding disease prevention and control of HuNoV infection, the most often correctly identified item was no commercial vaccine so far, with an accuracy of 79.89%. However, only 29.83% parents knew that the places contaminated by vomitus or stool should be cleaned and disinfected. Less than half of the respondents (49.42%) perceived that chlorine-containing disinfectants were the effective scavengers against HuNoV. About 50% of parents knew that virus-infected children should be kept in quarantine at home to avoid spreading the disease. Moreover, only 60.36% parents understood that antibiotics cannot be used to treat HuNoV infection. Encouragingly, most of parents knew that good food hygiene, adequate handwashing, and environmental cleaning can prevent HuNoV infection.

### 3.3. Analysis of Factors Affecting Knowledge about HuNoV

First, each independent variable was entered into bivariate logistic regression models to evaluate the association with sociodemographic characteristics. In pathogenic knowledge of HuNoV, variables including level of education, occupation, grade of children, history of infection, and HuNoV learning experience had significant associations (*p* < 0.05). Additionally, in the prevention and control of HuNoV, variables such as level of education, occupation, grade of children, types of kindergarten, and HuNoV learning experience were shown to be significant associations (*p* < 0.05). History of infection had no association with parents’ knowledge about prevention and control of HuNoV. Then, all the variables meeting the criteria of significance (*p* ≤ 0.2) were entered into a multivariate logistic regression model. The multivariate model revealed that level of education and HuNoV learning experience were shown to be significant predictors of pathogenic knowledge of HuNoV. Parents with a vocational degree, bachelor’s degree, and graduate degree were about 2.27, 3.75, and 3.32 times more likely to be knowledgeable than those with a high school and below educational level (AOR: 2.27, 95% CI: 1.13–4.57; AOR: 3.75, 95% CI: 1.77–6.63; AOR: 3.32, 95% CI: 1.49–6.80), respectively. Regarding HuNoV learning experience, participants who did not have experience in HuNoV learning were 41% less likely to be knowledgeable than parents who had participated in HuNoV learning before (AOR: 0.59, 95% CI: 0.19–1.30).

Concerning parents’ knowledge about prevention and control of HuNoV, the multivariable logistic regression analysis showed that variables including level of education, occupation, and types of kindergarten were predictors. As presented in Table 3, those with a vocational degree (AOR: 2.31, 95% CI: 1.18–4.54), bachelor’s degree (AOR: 3.25, 95% CI: 1.72–6.12), and graduate degree (AOR: 2.74, 95% CI: 1.31–5.76) educational statuses were significantly more likely to be knowledgeable. Parents who engaged in non-medical work were significantly less likely to have good knowledge about prevention and control of HuNoV. Parents whose children attended privately-owned kindergartens were 1.6 times more likely to have good knowledge about prevention and control of HuNoV (AOR: 1.60, 95% CI: 1.09–2.35) than those whose children attended government-owned kindergartens.

### 3.4. Source of Information and Knowledge Requirement about HuNoV

In order to evaluate the participants’ health information-seeking behavior, parents were asked about their willingness to learn about HuNoV; further, we investigated the common sources of information they used. Of 771 parents, 94.55% (*n* = 729) had strong desires to know about HuNoV. The most common source of HuNoV knowledge was the Internet (93.4%), followed by knowledge training sessions in kindergartens (48.9%), while much lower proportions of parents relied on community information (38.3%) and television (35.9%). Only 16.82% participants chose doctors/friends as their choice to obtain information. Meanwhile, we investigated their expected approaches to obtain information on HuNoV. As expected, the expected approaches to learn about HuNoV were the same as the common sources of information they used.

## 4. Discussion

China is one of the 15 countries with a high incidence of diarrhea [17]. As one of the causative agents of acute gastroenteritis, HuNoV infection is common among children. The increasing outbreaks of HuNoV infection in kindergartens and schools have become a major concern of public health and brought serious economic burden to families and society. Parents, as the main caregivers for children, pose an important role in children’s health and disease control. Previous studies revealed that low parental health literacy correlates with worse health outcomes in their children [25,26]. Good knowledge about virus infection in the general public will help to reduce the health consequences of virus infection and to increase the population’s willingness to pay for future vaccination [24,27]. This present study demonstrates that parents whose children attended kindergartens had poor awareness of HuNoV infection in Chengdu. Only 81.97% of respondents had heard about HuNoV, with an average awareness rate of 62.64%, which was lower than the awareness about measles (>70%) [28], hand foot and mouth disease (HFMD) (83%) [29], and COVID-19 among parents in China (>90%) [30]. Analogously, findings of teachers’ awareness about HuNoV infection in nursery centers and primary schools were also low, with a 48.42% cognitive rate [31]. This lower awareness about HuNoV may be due to the late discovery of HuNoV outbreaks in China and the lack of effective health education. It is worth noting that children with HuNoV infection history made up only 3.79%, which was much lower than the prevalence (20.5%) among children in Chengdu [11]. This may be because some of the infected children might not be diagnosed as HuNoV infection or due to recall bias of their parents.

From the respondents who reported hearing something specific about NoVs, several gaps in HuNoV infection knowledge were identified. First, most parents did not correctly perceive the duration and symptoms of illness, and nearly a third of the respondents were even unable to distinguish HuNoV infection from food poisoning. Although children were requested to perform daily morning checkups before entering kindergarten, the neglect or misdiagnosis of the HuNoV infection by parents may cause more infections in children. It was reported that several HuNoV outbreaks in kindergartens might be attributed to neglected or asymptomatic HuNoV infection in children [32,33,34]. The second gap was the seasonal infection of HuNoV. Only half of the respondents correctly answered that most HuNoV outbreaks occur in winter, which was consistent with one gap in food-safety professionals’ knowledge about HuNoV [35]. HuNoV outbreaks occur all year round but are more frequent in the winter. Moreover, nearly 40% of parents had the misperception whereby, once infected with HuNoV, a person would never become infected with the virus again. This suggests that most parents had limited pathogenic knowledge of HuNoV.

With regards to the gaps in knowledge about prevention and control of HuNoV, less than 30% of parents were correctly aware that the areas contaminated by suspected vomitus or stool should be immediately cleaned and disinfected with chlorine-containing disinfectants. Previous HuNoV outbreaks revealed that HuNoV could also be transmitted via airborne transmission mode, particularly in closed settings. The right way to clean and disinfect the vomitus or stool is initial cleaning to remove soiling followed by using chlorine-containing disinfectants to disinfect against HuNoVs [36]. Likewise, almost 50% of parents did not know that alcohol is less effective against HuNoV, suggesting that parents had a lack of knowledge on prevention of HuNoV. Moreover, approximately half of respondents were unaware that children infected with HuNoV should be kept in quarantine at home. Moreover, unexpectedly, nearly 40% of parents misperceived that antibiotics can be used to treat HuNoV infection. It was understood that most of the investigated kindergartens had given 1–3 training sessions on knowledge of infectious diseases to parents every semester, including measles, flu, mumps, HFMD, and HuNoV infection [23,37]. However, only 45.7% of parents (289/632) responded that they had experience in HuNoV learning, and 19.2% of respondents had never even heard about the virus, which suggests that more efforts are needed to inform parents about HuNoV infection.

Factors associated with good knowledge regarding HuNoV infection were level of education, occupation, history of infection, and HuNoV learning experience. In terms of level of education, parents who had a lower level of education were significantly less knowledgeable about HuNoV infection, which is similar to the findings relative to consumers’ knowledge about norovirus prevention and control [38]. As previous surveys have shown, health literacy was associated with a higher level of education [24,39]. Nevertheless, in order to prevent HuNoV infection, we need to educate parents regardless of their educational background. In terms of occupation, parents who engaged in medical work were significantly more likely to have good knowledge about HuNoV, but there was no significant difference between teacher and other jobs. It is not difficult to understand that parents who engaged in medical work had more experience in the matter or had chances to come into contact with HuNoV knowledge. As for the occupation of teacher, originally, we hypothesized that parents who were teachers may have had more chances to learn about HuNoV due to the regular infectious disease training sessions held in kindergartens or schools. However, the findings show that parents who were teachers were 61% less likely to be knowledgeable than parents who engaged in medical work. This suggests that parents who were teachers had poor awareness of HuNoV infection, and special attention should be paid to this group of parents to promote awareness about HuNoV infection. As for the factors of history of infection and HuNoV learning experience, it is not difficult to understand that this group of parents have good knowledge about HuNoV. Parents whose children had a history of infection and who had HuNoV learning experience were significantly more likely to be knowledgeable. This implies that HuNoV knowledge learning plays a vital role in promoting awareness about HuNoV infection and prevention and control of HuNoV infection. It is interesting that parents whose children attended privately-owned kindergartens were more likely to have good knowledge about prevention and control of HuNoV. One possible reason is that more parents (*n* = 197) whose children attended privately-owned kindergartens had HuNoV learning experience compared with the parents of children from public-owned kindergartens (*n* = 92).

Generally, people’s health knowledge depends on various factors, including the sources of the information they use [24]. Our results show that the Internet was the most common source of health information by parents, and 91.1% of respondents preferred to obtain health information from the Internet. The second popular source was knowledge training sessions in kindergartens. This was consistent with the results that the Internet was also the most common source of information from which American consumers obtain norovirus information [38]. Significantly, the potential concerns with the use of Internet information are whether it is easily understandable or its sources reliable. Thus, it is important for sources, such as government organizations or hospitals, to provide accurate knowledge about HuNoV [38].

The study had several limitations. First, the participants in this study may not represent the whole parent population in Chengdu. The percentage of children who had a history of HuNoV infection in this study, for instance, was much lower than the norovirus-positive rate reported in hospitalized children in Chengdu [11]. Second, a self-administered questionnaire was used in this study; therefore, the accuracy of the survey data relied on the parents’ honesty, memories, and ability to understand the questions correctly. Other limitations include possible response bias due to the parent’s recent participation in knowledge training sessions, which provided education about noroviruses. Additionally, a minor weakness that should be acknowledged is the reliance on recall of symptoms. Nevertheless, the present study provides an overall view of parents’ awareness of HuNoV infection, and this gives a reference for the design of similar studies among parents or other groups in other parts of China.

In summary, our work provides insights into the parents’ awareness of HuNoV infection and the sociodemographic factors affecting their knowledge. The parents’ level of awareness about HuNoV infection was suboptimal. Sociodemographic characteristics, such as level of education, occupation, history of infection, and HuNoV learning experience, were important factors associated with parents’ knowledge of HuNoV infection. Given the high prevalence rate of HuNoV infection in children under 5 years old, more efforts and resources must be devoted to educating parents and children on HuNoV infection prevention. The findings from our study provide knowledge areas and educational materials for improvement.

## Figures and Tables

**Table 1 ijerph-19-01570-t001:** Sociodemographic characteristics of parents (*n* = 632) and their association with knowledge score.

Characteristics of Parents	*n*	Pathogenic Knowledge Score		Prevention and Control Score	
		Mean ± SD	*p*-Value	Mean ± SD	*p*-Value
**Level of Education**					
High school degree and below	50	7.28 ± 1.98	<0.001	6.84 ± 1.61	<0.001
Vocational degree	136	8.14 ± 1.55		7.57 ± 1.69	
Bachelor’s degree	308	8.52 ± 1.50		7.92 ± 1.53	
Graduate degree	138	8.51 ± 1.77		7.70 ± 1.71	
**Occupation**					
Medical staff	35	9.11 ± 1.28	0.01	8.71 ± 1.34	<0.001
Teacher	128	8.37 ± 1.68		7.77 ± 1.52	
Others	469	8.27± 1.65		7.62 ± 1.66	
**Grade of children**					
Junior class	203	8.43 ± 1.60	0.38	7.72 ± 1.65	0.33
Middle class	250	8.35 ± 1.57		7.81 ± 1.59	
Top class	179	8.20 ± 1.80		7.57 ± 1.68	
**Types of kindergarten**					
Government owned	221	8.28 ± 1.87	0.53	7.63 ± 1.79	0.40
Privately owned	411	8.37 ± 1.51		7.75 ± 1.55	
**History of infection**					
Yes	24	8.75 ± 0.85	0.03	8.21 ± 1.56	0.13
No	608	8.32 ± 1.67		7.69 ± 1.64	
**HuNoV learning experience**					
Yes	289	8.62 ± 1.43	<0.001	7.98 ± 1.45	<0.001
No	343	8.09 ± 1.77		7.49 ± 1.75	

**Table 2 ijerph-19-01570-t002:** Parents’ awareness about pathogenic knowledge of HuNoV, disease prevention, and control.

Questions	Number of Parents (*n*)	**Correctly Answered (%)**
**Pathogenic Knowledge of HuNoV**		**62.12%**
If a child is diagnosed with HuNoV infection, infection may spread to other people.	578	74.97%
The incubation period of HuNoV infection is usually from 24 to 48 h.	345	44.75%
What are the common signs of HuNoV infection in children?	605	78.47%
There is no difference between norovirus infection and food poisoning for similar symptoms.	497	64.46%
The symptoms of HuNoV infection in children are generally relieved within 2–3 days.	194	25.16%
Are kindergartens high-risk places for HuNoV infection outbreaks?	551	71.47%
Children can be infected with HuNoV by contacting a virus carrier or shellfish.	536	69.52%
What is the most common mode of transmission for HuNoV?	581	75.36%
Once infected with HuNoV, a person will never become infected by the virus again.	474	61.48%
Children younger than 5 years old and older are more susceptible to severe HuNoV infections.	606	78.59%
In which season do most HuNoV outbreaks occur?	416	53.96%
**Disease prevention and control**		**63.22%**
Mild cases of children infected with HuNoV can be nursed at home.	517	67.06%
If a child infected with HuNoV, does he/she need to be kept in quarantine?	391	50.71%
If a child is infected with HuNoV, how long should he/she stay at home even after he/she no longer shows symptoms of illness?	607	78.73%
The areas contaminated by vomitus or stool should be immediately cleaned and disinfected with chlorine-containing disinfectants.	230	29.83%
Alcohol is as effective against HuNoV as chlorine-containing disinfectants.	381	49.42%
There is currently no available vaccine against HuNoV.	616	79.89%
Antibiotics cannot be used to treat HuNoV infection.	465	60.36%
Adequate handwashing can prevent viral infection.	586	75.87%
Adequate environmental cleaning can prevent HuNoV from infecting others.	596	77.30%
Good food hygiene, including cooking raw shellfish and keeping raw food separate from cooked food, can prevent virus infection.	587	76.13%

**Table 3 ijerph-19-01570-t003:** Odds ratios of knowledge scores with socio-demographic characteristics (good vs. poor).

Variables	Pathogenic Knowledge				Prevention and Control			
	COR (95% CI)	*p*	AOR (95% CI)	*p*	COR (95% CI)	*p*	AOR (95% CI)	*p*
**Level of education**		0.01		<0.001		0.04		<0.001
High school degree and below	1		1		1		1	
Vocational degree	2.40 (1.20–4.80)		2.27 (1.13–4.57)		2.46 (1.26–4.82)		2.31 (1.18–4.54)	
Bachelor’s degree	3.75 (1.97–7.18)		3.43 (1.77–6.63)		3.64 (1.95–6.80)		3.25 (1.72–6.12)	
Graduate degree	3.32 (1.66–6.63)		3.19 (1.49–6.80)		3.13 (1.60–6.14)		2.74 (1.31–5.76)	
**Occupation**		<0.001		0.19		<0.001		0.01
Medical staff	1		1		1		1	
Teacher	0.39 (0.17–0.93)		0.35 (0.14–0.86)		0.24 (0.08–0.72)		0.23 (0.07–0.70)	
Others	0.35 (0.16–0.79)		0.41 (0.18–0.95)		0.19 (0.65–0.54)		0.21 (0.07–0.61)	
**Grade of children**		0.03		0.71		<0.001		0.76
Junior class	1		1		1		1	
Middle class	0.85 (0.59–1.23)		0.89 (0.60–1.30)		0.98 (0.67–1.43)		1.04 (0.70–1.54)	
Top class	1.02 (0.68–1.53)		1.10 (0.72–1.67)		0.95 (0.63–1.44)		1.07 (0.70–1.64)	
**Types of kindergarten**		0.12		0.84		<0.001		0.02
Government owned	1		1		1		1	
Privately owned	1.06 (0.76–1.48)		1.39 (0.96–2.03)		1.13 (0.81–1.58)		1.60 (1.09–2.35)	
**History of Infection**		0.02		0.16		0.05		0.56
Yes	1		1		1		1	
No	0.42 (0.16–1.06)		0.50 (0.19–1.30)		0.66 (0.27–1.62)		0.76 (0.30–1.91)	
**HuNoV learning** **experience**		<0.001		<0.001		<0.001		0.08
Yes	1		1		1		1	
No	0.57 (0.41–0.78)		0.59 (0.42–0.81)		0.71 (0.51–0.98)		0.74 (0.53–1.04)	

## Data Availability

The data presented in this study are available upon request from the corresponding author.

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
