# Peer review of "Knowledge, Awareness, and Prevention of Norovirus Infection among Kindergarten Parents in Chengdu, China"

_ijerph, 2022, doi:10.3390/ijerph19031570_

Round 1

Reviewer 1 Report

The authors have studied the knowledge and awareness of norovirus gastroenteritis among parents of kindergarten children. The parents completed an online questionnaire about several aspects of norovirus. This study presents interesting results and leads for improvement of knowledge and prevention of norovirus gastroenteritis. Nevertheless, the manuscript needs extensive editing of English language. Other comments:

  1. Introduction, lines 36-38. Please consider removing ‘As a typical foodborne virus’ as it is irrelevant in the context of this sentence.
  2. Material and Methods, lines 92-93. Please specify the (mean) ages of the children in the three different grades.
  3. Material and Methods, lines 112-113. What is meant by ‘experience HuNoV learning’? Please give a definition or explanation.
  4. Material and Methods, line 120. Is this cutoff value calculated as 80% of the parents or on 80% of the scores?
  5. Results, line 171, ‘nearly 3.79%’. This is a strange combination, I would suggest to change it into ‘3.79%’ or ‘nearly 4%’.
  6. Table 1 & 3. A p-value of 0.00 is confusing. Please consider changing this into something like ‘<0.01’ or ‘<0.001’.
  7. Results, lines 212-214. ‘But’ is inappropriate here, in my view. I would suggest to mention here the similarities and differences in variables found in the knowledge part.
  8. Results, line 242. I think ‘smartphone’ can be left out here, as I reckon that the smartphone is used to go to internet for the information, thus internet being the source and not the smartphone itself.
  9. Results, lines 245-248. I don’t understand what the difference of these sentences is compared to the sentences above (lines 241-245). Please rephrase if something else is meant or otherwise remove.
  10. Discussion, lines 260-263. Could you please add the percentages of awareness of the diseases mentioned for comparison with the percentage found for norovirus in the present study?

Author Response

Re: Revised Manuscript ijerph-1549809

On behalf of all co-authors, I would like to thank you and the reviewers for favorable comments and constructive suggestions on the manuscript (MS) ijerph-1549809. The Reviewers has been very kind and generous, and provided valuable comments and detailed suggestions for us to improve the quality of the MS, for which we are very grateful. According to the reviewersËŠ comments, we have tried best to modify the manuscript to meet with the requirements of the journal. We have provided revised version of our MS in “Tracked Changes” file and “Clean” one. In the following, we detail our point-by-point responses to these specific comments and suggestions.

Responses to comments and suggestions of Reviewer #1:

Point 1: The authors have studied the knowledge and awareness of norovirus gastroenteritis among parents of kindergarten children. The parents completed an online questionnaire about several aspects of norovirus. This study presents interesting results and leads for improvement of knowledge and prevention of norovirus gastroenteritis. Nevertheless, the manuscript needs extensive editing of English language.

Response: We thank the reviewer #1 very much for his/her favorable and positive comments on our MS. We have entrusted MDPI to improve the language of the MS. We would like to acknowledge MDPI for English language editing.

Point 2: Introduction, lines 36-38. Please consider removing ‘As a typical foodborne virus’ as it is irrelevant in the context of this sentence.

Response: Good suggestions. We removed the ‘As a typical foodborne virus’.

Point 3: Material and Methods, lines 92-93. Please specify the (mean) ages of the children in the three different grades.

Response: Thanks for your constructive suggestions. We have re-counted the ages of the children and specified accordingly.

Point 4: Material and Methods, lines 112-113. What is meant by ‘experience HuNoV learning’? Please give a definition or explanation.

Response: We have changed ‘experience HuNoV learning’ as ‘HuNoV learning experience’. That means parents who had participated in knowledge training sessions held by kindergartens and/or communities.

Point 5: Material and Methods, line 120. Is this cutoff value calculated as 80% of the parents or on 80% of the scores?

Response: Thank you for your valuable suggestions. We have revised as “…. based on an 80% cutoff value of the scores”.

Point 6: Results, line 171, ‘nearly 3.79%’. This is a strange combination; I would suggest to change it into ‘3.79%’ or ‘nearly 4%’.

Response: Good suggestions. We have removed “nearly”.

Point 7: Table 1 & 3. A p-value of 0.00 is confusing. Please consider changing this into something like ‘<0.01’ or ‘<0.001’.

Response: Revised accordingly.

Point 8: Results, lines 212-214. ‘But’ is inappropriate here, in my view. I would suggest to mention here the similarities and differences in variables found in the knowledge part.

Response: Good suggestions. We have revised accordingly.

Point 9: Results, line 242. I think ‘smartphone’ can be left out here, as I reckon that the smartphone is used to go to internet for the information, thus internet being the source and not the smartphone itself.

Response: Good suggestions. We deleted the ‘smartphone’.

Point 10: Results, lines 245-248. I don’t understand what the difference of these sentences is compared to the sentences above (lines 241-245). Please rephrase if something else is meant or otherwise remove.

Response: Good suggestions. We have improved accordingly.

Point 11: Discussion, lines 260-263. Could you please add the percentages of awareness of the diseases mentioned for comparison with the percentage found for norovirus in the present study?

Response: We have added the percentages of awareness of the diseases mentioned in the MS.

We sincerely hope that the MS has been revised satisfactorily, and it meets the standard for publication in the journal.

Reviewer 2 Report

  1. 1 lines 31-37 words with various letter sizes

p.2 line 55-58- wording of this sentence is confusing

p.2 line 80-81- wording of this sentence is confusing

p.2 line 59- Add more references

p.2 line 91, section 2.1: Please provide the age of the participants recruited for the study.

p.3 line 101- sentence needs restructuring

  1. 6 Table 3 line 236 methods do not describe the definition of good vs poor.

How do authors explain the low proportion of knowledge about prevention and control of HuNoV attending government owned kindergarten

A minor weakness that should be acknowledged in the reliance on recall of symptoms

In Table 2, please include “n”

Author Response

Re: Revised Manuscript ijerph-1549809

On behalf of all co-authors, I would like to thank you and the reviewers for favorable comments and constructive suggestions on the manuscript (MS) ijerph-1549809. The Reviewers has been very kind and generous, and provided valuable comments and detailed suggestions for us to improve the quality of the MS, for which we are very grateful. According to the reviewersËŠ comments, we have tried best to modify the manuscript to meet with the requirements of the journal. We have provided revised version of our MS in “Tracked Changes” file and “Clean” one. In the following, we detail our point-by-point responses to these specific comments and suggestions.

Responses to comments and suggestions of Reviewer #2:

Point 1: lines 31-37 words with various letter sizes.

Response: Revised accordingly.

Point 2: p.2 line 55-58- wording of this sentence is confusing

Response: We have rephrased this sentence.

Point 3: p.2 line 80-81- wording of this sentence is confusing

Response: We have rephrased this sentence.

Point 4: p.2 line 59- Add more references

Response: Good suggestion. We have added three references.

Point 5: p.2 line 91, section 2.1: Please provide the age of the participants recruited for the study.

Response: Thanks for your constructive suggestions. We have re-counted the ages of the children and specified accordingly.

Point 6: p.3 line 101- sentence needs restructuring.

Response: Revised accordingly.

Point 7: Table 3 line 236 methods do not describe the definition of good vs poor.

Response: Thanks for your constructive comments. The definitions of “good” or “poor” were contained in p.3 line128-129.

Point 8: How do authors explain the low proportion of knowledge about prevention and control of HuNoV attending government owned kindergarten?

Response: Revised accordingly.

Point 9: A minor weakness that should be acknowledged in the reliance on recall of symptoms.

Response: Thanks for your good suggestions. We have added this point in p.9 line 374.

Point 10: In Table 2, please include “n”

Response: We have added the number of parents (n) into Table 2.

We sincerely hope that the MS has been revised satisfactorily, and it meets the standard for publication in the journal.

Reviewer 3 Report

This is a well design and conducted study about public knowledge of human Norovirus. It focused on the parents of kindergartener in Chengdu, China. The information from this study is helpful to improve the awareness and prevention of Norovirus infection.

The English writing in this manuscript should be improved. Some words and descriptions may not be accurate. Restrict review and modification are necessary.

Some comments in detail:

  1. The font size in first paragraph of Introduction is not identical. Please correct it.
  2. Line 166-169: The education background was specified as postgraduate, graduate, junior college and secondary school. Is junior college similar with short term college? Is secondary school similar with community college? It would be good to specify each level a little more clear.
  3. The children with Norovirus infection history was only 3.79% in the studied pool. This rate is too low. Some of the infected kids might not be diagnosed as Norovirus infection.
  4. Table 1, 3: Some of the P value were defined as 0.00. Is it just less than 0.001 or 0? The real P value should be provided.
  5. Line 212: The word “but” is not proper because both sentences here are significant associated (P<0.05).
  6. Line 304: “without regard to” could be changed to “regardless”. There are more similar problems. I can’t list them all. So please proofread the manuscript carefully.
  7. Line 322: The word “literacy” could be changed to “knowledge”.

Author Response

Re: Revised Manuscript ijerph-1549809

On behalf of all co-authors, I would like to thank you and the reviewers for favorable comments and constructive suggestions on the manuscript (MS) ijerph-1549809. The Reviewers has been very kind and generous, and provided valuable comments and detailed suggestions for us to improve the quality of the MS, for which we are very grateful. According to the reviewersËŠ comments, we have tried best to modify the manuscript to meet with the requirements of the journal. We have provided revised version of our MS in “Tracked Changes” file and “Clean” one. In the following, we detail our point-by-point responses to these specific comments and suggestions.

Responses to comments and suggestions of Reviewer #3:

Point 1: This is a well design and conducted study about public knowledge of human Norovirus. It focused on the parents of kindergartener in Chengdu, China. The information from this study is helpful to improve the awareness and prevention of Norovirus infection. The English writing in this manuscript should be improved. Some words and descriptions may not be accurate. Restrict review and modification are necessary.

Response: We thank the reviewer #3 very much for his/her favorable and positive comments on our MS. We have entrusted MDPI to improve the language of the MS. We would like to acknowledge MDPI for English language editing.

Point 2: The font size in first paragraph of Introduction is not identical. Please correct it.

Response: We have revised accordingly.

Point 3: Line 166-169: The education background was specified as postgraduate, graduate, junior college and secondary school. Is junior college similar with short term college? Is secondary school similar with community college? It would be good to specify each level a little more clear.

Response: We have specified accordingly.

Point 4: The children with Norovirus infection history was only 3.79% in the studied pool. This rate is too low. Some of the infected kids might not be diagnosed as Norovirus infection.

Response: Good suggestion. We have mentioned this in the Discussion part.

Point 5: Table 1, 3: Some of the P value were defined as 0.00. Is it just less than 0.001 or 0? The real P value should be provided.

Response: Thanks for your suggestion. We have provided.

Point 6: Line 212: The word “but” is not proper because both sentences here are significant associated (P<0.05).

Response: Revised accordingly.

Point 7: Line 304: “without regard to” could be changed to “regardless”. There are more similar problems. I can’t list them all. So please proofread the manuscript carefully.

Response: Thanks for your good suggestions and generous comments. We have revised and proofread the whole text.

Point 8: Line 322: The word “literacy” could be changed to “knowledge”.

Response: Revised accordingly.

We sincerely hope that the MS has been revised satisfactorily, and it meets the standard for publication in the journal.

This manuscript is a resubmission of an earlier submission. The following is a list of the peer review reports and author responses from that submission.